# HaarPooling: Graph Pooling with Compressive Haar Basis

## Abstract

Deep Graph Neural Networks (GNNs) are instrumental in graph classification and graph-based regression tasks. In these tasks, graph pooling is a critical ingredient by which GNNs adapt to input graphs of varying size and structure. We propose a new graph pooling operation based on compressive Haar transforms, called *HaarPooling*. HaarPooling is computed by following a chain of sequential clusterings of the input graph. The input of each pooling layer is transformed by the compressive Haar basis of the corresponding clustering. HaarPooling operates in the frequency domain by the synthesis of nodes in the same cluster and filters out fine detail information by compressive Haar transforms. Such transforms provide an effective characterization of the data and preserve the structure information of the input graph. By the sparsity of the Haar basis, the computation of HaarPooling is of linear complexity. The GNN with HaarPooling and existing graph convolution layers achieves state-of-the-art performance on diverse graph classification problems.

## 1 Introduction

Graph Neural Networks (GNNs) have demonstrated excellent performance in node classification tasks and are very promising in graph classification and regression (Bronstein et al., 2017; Battaglia et al., 2018; Zhang et al., 2018b; Zhou et al., 2018; Wu et al., 2019). In node classification, the input is a single graph with missing node labels that are to be predicted from the known node labels. In this problem, GNNs with appropriate graph convolutions can be trained based on the single graph that is provided, and achieve state-of-the-art performance (Defferrard et al., 2016; Kipf & Welling, 2017; Ma et al., 2019b). Different from node classification, graph classification is a task where the label of any given graph-structured sample is to be predicted based on a training set of labeled graph-structured samples. This is similar to the image classification task tackled by traditional deep convolutional neural networks. The major difference is that here each input sample may have an arbitrary adjacency structure, instead of the fixed regular grids that are used in images. An example of graph-structured data are the molecules of different sizes shown in Figure 1. This raises two important challenges: 1) How can GNNs exploit the graph structure information of the input data? 2) How can GNNs handle input graphs with varying number of nodes and connectivity structures?

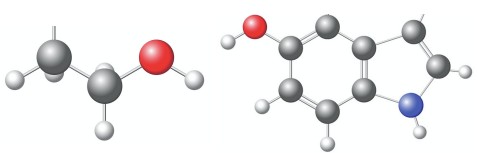

Figure 1: Graph-structured data: two molecules with atoms as nodes and bonds as edges. Each molecule has a different number of nodes and molecular structure. In graph classification (and regression), each input datum is an individual graph with features defined on the graph nodes (e.g., indicating the chemical element).

These problems have motivated the design of proper *graph convolution* and *graph pooling* to allow GNNs to capture the geometric information of each data sample (Zhang et al., 2018a; Ying et al., 2018; Cangea et al., 2018; Gao & Ji, 2019; Knyazev et al., 2019; Ma et al., 2019a; Lee et al., 2019). Graph convolution plays an important role especially in question 1).

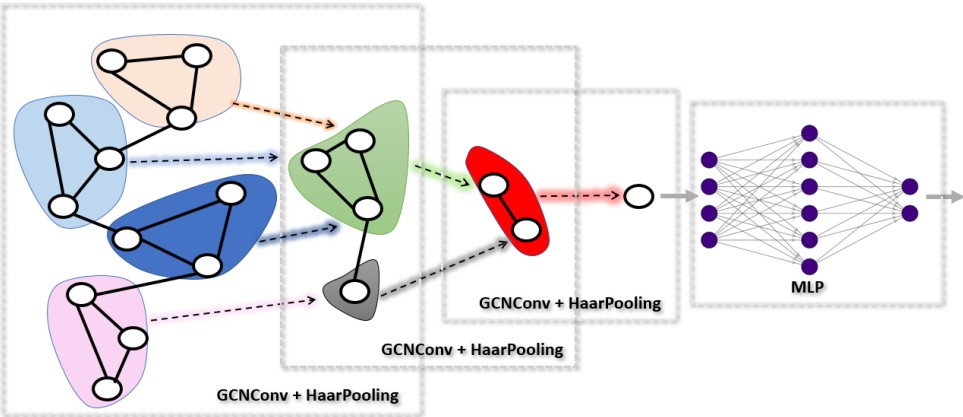

Figure 2: Computational flow of a Graph Neural Network consisting of three blocks of GCN graph convolutional and HaarPooling layers, followed by an MLP. In this example, the output feature of the last pooling layer has dimension 4, which is the number of input units of the MLP.

The following graph convolution, as proposed by Kipf & Welling (2017), is a widely accepted example:

$$X^{\text{out}} = \widehat{A} X^{\text{in}} W. \tag{1}$$

Here $\widehat{A} = \widetilde{D}^{-1/2}(A + I)\widetilde{D}^{-1/2} \in \mathbb{R}^{N \times N}$ is a normalized version of the adjacency matrix $A$ of the input graph, where $I$ is the identity matrix and $\widetilde{D}$ is the degree diagonal matrix for $A + I$. Further, $X^{\text{in}} \in \mathbb{R}^{N \times d}$ is the array of $d$-dimensional features on the $N$ nodes of the graph, and $W \in \mathbb{R}^{d \times m}$ is the filter parameter matrix. The graph convolution in equation 1 captures the structural information of the input in terms of $A$ (or $\widehat{A}$), and $W$ transforms the feature dimension from $d$ to $m$. The filter size $d \times m$ does not depend on the graph size, which allows a fixed network architecture to process input graphs of varying size. However, the GCN convolution preserves the number of nodes and hence the output dimension of the network is not unique. Graph pooling provides an effective way to overcome this obstacle. Among several approaches that have been proposed, only EigenPooling (Ma et al., 2019a) incorporates both features and graph structure. However, this is based on eigenpairs of graph Laplacian and suffers from a high computational cost. We provide an overview of this and other graph pooling methods in Section 2.

In this paper, we propose a new graph pooling strategy based on a sparse Haar representation of the data, which we call *HaarPooling*. This is based on the *Haar basis* (Li et al., 2019), which incorporates graph structure and features, and is computationally efficient. Suppose we have an input $X^{\text{in}} \in \mathbb{R}^{N \times d}$. HaarPooling is then defined as

$$X^{\text{out}} = \Phi^T X^{\text{in}}, \tag{2}$$

where $\Phi \in \mathbb{R}^{N \times N_1}$, $N_1 < N$. Each column of $\Phi$ is a *compressive Haar basis* vector. By applying HaarPooling in equation 2, the number of nodes is compressed from $N$ to $N_1$. The Haar basis provides a sparse representation which distills graph structural information. Cascading pooling layers we can obtain an output of a fixed dimension, regardless of the size of the inputs. The sparsity of the Haar basis matrix ensures that the computation of HaarPooling is efficient. Generating the Haar basis and computing the Haar transform has a computational cost $\mathcal{O}(N)$ (up to a log term of $N$) for input graphs with $N$ nodes. Experiments in Section 5 demonstrate that GNNs with HaarPooling achieve state-of-the-art performance on various graph classification tasks.

This paper is organized as follows. Section 2 gives an overview of existing work on graph pooling. Section 3 details the components and computational flow for HaarPooling. Section 4 provides the mathematical details on HaarPooling, including the compressive Haar basis, compressive Haar transforms, and efficient implementations. Section 5 reports our experimental results on benchmark graph classification tasks compared with existing graph pooling methods. Section 6 concludes the paper. Proofs and implementation details are deferred to the appendix.

## 2 RELATED WORK

Graph pooling is a necessary step when building a GNN model for graph classification, as one needs a unified graph-level rather than node-level representation for graph inputs for which size and topology are changing. The most direct pooling method takes the global mean and sum of node representations obtained by the graph convolutional layer (Duvenaud et al., 2015) as a simple graph-level representation. However, this pooling operation treats all the nodes equally and ignores the global geometry of the graph. ChebNet (Defferrard et al., 2016) uses a graph coarsening procedure to build the pooling module, which requires graph clustering algorithms to obtain subgraphs. One drawback of this topology-based strategy is that it does not incorporate the node features in the pooling. Global pooling methods consider the information of node embeddings to obtain the entire graph representation. As a general framework for graph classification problems, MPNN (Gilmer et al., 2017) uses the Set2Set method (Vinyals et al., 2015) to obtain graph-level representations. Zhang et al. (2018a) proposed a SortPool method that sorts feature representation of nodes before feeding them into traditional 1-D convolutional and dense layers. But these global pooling techniques cannot guarantee hierarchical graph representations that may contain useful information in the graph structure. A prominent recent idea is to build a differentiable and data-dependent pooling layer with learnable operations/parameters, which has brought substantial improvements on graph classification tasks. Ying et al. (2018) proposed a differentiable pooling layer (DiffPool) that learns a cluster assignment matrix over the nodes using the output of a GNN model. One common problem with DiffPool is its huge storage complexity, which results from the computation of the soft clustering assignments. Cangea et al. (2018); Gao & Ji (2019); Knyazev et al. (2019) used a Top-K pooling method that samples a subset of important nodes by employing a trainable projection vector. Lee et al. (2019) introduced Self-Attention Graph Pooling (SAGPool) by replacing the way node scores are computed in Top-K pooling by a GCN module. These hierarchical pooling methods technically still employ mean/max pooling procedures to aggregate the feature representation of super-nodes, which would lead to information loss. Diehl et al. (2019) proposed EdgePool which is a scheme which considers edge contraction and thus takes into account of graph structure in pooling.

There are also spectral-based pooling methods that take account of both the graph structure and its node features. Noutahi et al. (2019) proposed the Laplacian Pooling (LaPool) method that dynamically selects centroid nodes and their corresponding follower nodes by an attention mechanism that uses the graph Laplacian. Ma et al. (2019a) introduced EigenPool which uses local graph Fourier transform to extract subgraph information utilizing both node features and structure of the subgraph. Its potential drawback lies in the inherent bottleneck of computing Laplacian-based graph Fourier transform, given the huge cost in the eigendecomposition of the graph Laplacian. This shortcoming partially motivates our present work.

## 3 HAARPOOLING

In this section we give an overview of the proposed HaarPooling framework. First we define the pooling architecture in terms of a chain, i.e., a sequence of graphs $(\mathcal{G}_0, \mathcal{G}_1, \ldots, \mathcal{G}_K)$, where the nodes of each $\mathcal{G}_{j+1}$ correspond to clusters of nodes of $\mathcal{G}_{j+1}$, $j = 0, \ldots, K - 1$. Each layer in the chain determines which sets of nodes are pooled together. Then we construct the compressive Haar transform, which compresses the dimension of the features.

**Chain of coarse-grained graphs for pooling**   Graph pooling amounts to defining a sequence of coarse-grained graphs. In our chain, each graph is an induced graph that arises from grouping (clustering) certain subsets of nodes from the previous graph. We use clustering algorithms to generate the groupings of nodes. There are many good candidates, such as spectral clustering (Shi & Malik, 2000), $k$-means clustering (Pakhira, 2014), DBSCAN (Ester et al., 1996), OPTICS (Ankerst et al., 1999) and METIS (Karypis & Kumar, 1998). Any of these will work with HaarPooling.

Figure 3 shows an example of a chain with 3 levels, for an input graph $\mathcal{G}_0$.

**Compressive Haar transforms on chain**   For each layer of the chain, we will have a feature representation. We define these in terms of the Haar basis. Haar basis represents graph-structured data by low and high frequency Haar coefficients in frequency domain. The low frequency coefficients con-

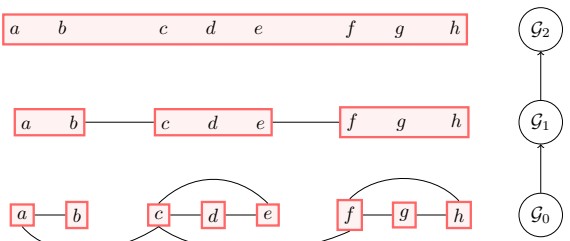

Figure 3: A coarse-grained chain of graphs, where the input has 8 nodes, the second level has 3 nodes, and the top level has single node.

tain the coarse information of the original data while the high frequency coefficients contain the fine details. In the HaarPooing, the data is pooled (or compressed) by discarding fine detail information.

The Haar basis can be compressed in each layer. Consider a chain where at level $j$ the two subsequent graphs have $N_{j+1}$ and $N_j$ nodes, $N_{j+1} < N_j$. For each of these graphs, we can create a Haar basis with $N_{j+1}$ and $N_j$ elements, respectively. The elements of the smaller layer are obtained by compressing a subset of the elements from the other layer. These new vectors form the matrix $\Phi_j$ of size $N_{j+1} \times N_j$. We call $\Phi_j$ *compressive Haar basis matrix* for this particular $j$th layer. This then defines the *compressive Haar transform* $\Phi_j^T X^{\text{in}}$ for feature $X^{\text{in}}$ with size $N_j \times d$.

**Computational strategy of HaarPooling**   The HaarPooling is then defined as follows.

**Definition 1** (HaarPooling). *The HaarPooling for a graph neural network with $K$ pooling layers is defined as*
$$X_j^{\text{out}} = \Phi_j^T X_j^{\text{in}}, \quad j = 0, 1, \dots, K-1,$$
*where $N_j > N_{j+1}$ and $N_K = 1$, $\Phi_j$ or $\Phi_{N_j \times N_{j+1}}^{(j)}$ is the $N_j \times N_{j+1}$ compressive Haar basis matrix for the $j$th layer, $X_j^{\text{in}} \in \mathbb{R}^{N_j \times d_j}$ is the input feature array, and $X_j^{\text{out}} \in \mathbb{R}^{N_{j+1} \times d_j}$ is the output feature array. The corresponding layer is called HaarPooling layer.*

HaarPooling has following key properties.

- The HaarPooling reduces layer by layer the first dimension of input feature. In the last pooling layer, the output feature is compressed as a vector with length $d_{K-1}$ and each original input sample would generate such a vector with the same length. This then makes it possible to deal with input graph-structured data with different size and structure.

- The HaarPooling uses the sparse Haar representation on chain structure. In each HaarPooling layer, the representation then combines the features of input $X_j^{\text{in}}$ with the geometric information of the graphs of the $j$th and $(j+1)$th layers of the chain.

- By the property of Haar basis, the HaarPooling only drops the high frequency (or detailed) information of the input data. The $X_j^{\text{out}}$ has good approximation to $X_j^{\text{in}}$. Thus, the major data information (i.e. the low frequency coefficients) is preserved in the pooling, and the loss of the information is small.

- Since the Haar basis matrix is very sparse, HaarPooling can be computed very fast, with near linear computational complexity.

**Example**   Figure 4 shows the computational details of the HaarPooling associated with the chain from Figure 3. There are two HaarPooling layers. In the first layer, the input $X_1^{\text{in}}$ with size $8 \times d_1$ is transformed by the compressive Haar basis matrix $\Phi_{8 \times 3}^{(0)}$ which consists of the first three column vectors of the full Haar basis $\Phi_{8 \times 8}^{(0)}$ in (a), and output is a $3 \times d_1$ matrix $X_1^{\text{out}}$. In the second layer, the input $X_2^{\text{in}}$ with size $3 \times d_2$ (usually $X_1^{\text{out}}$ followed by convolution) is transformed by the compressive Haar matrix $\Phi_{3 \times 1}^{(1)}$ which is the first column vector of the full Haar basis matrix $\Phi_{3 \times 3}^{(1)}$ in (b). By the construction of the Haar basis in relation to the chain (details in Appendix B), each of the first three column vectors $\phi_1^{(0)}, \phi_2^{(0)}$ and $\phi_3^{(0)}$ of $\Phi_{8 \times 3}^{(0)}$ has only up to three different values. This bound is exactly the number of nodes of $\mathcal{G}_1$. For each column $\phi_\ell^{(0)}$, all nodes with the same parent take the same value. Similarly, the $3 \times 1$ vector $\phi_1^{(1)}$ is constant. This means that the HaarPooling

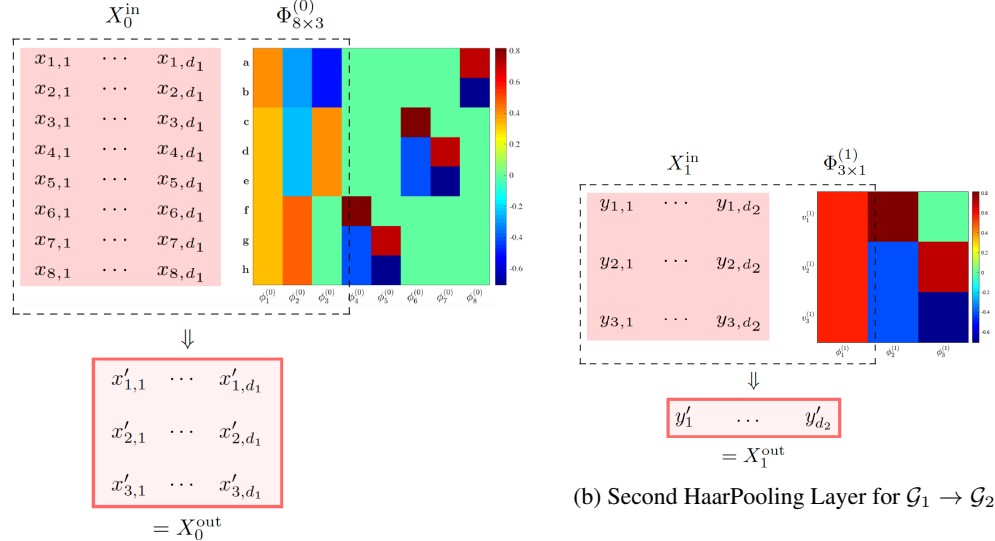

(a) First HaarPooling Layer for $\mathcal{G}_0 \to \mathcal{G}_1$

(b) Second HaarPooling Layer for $\mathcal{G}_1 \to \mathcal{G}_2$

Figure 4: Computational strategy of HaarPooling. We use the chain in Figure 3 and then there are two HaarPooling layers in the network from $\mathcal{G}_0 \to \mathcal{G}_1$ and $\mathcal{G}_1 \to \mathcal{G}_2$ respectively. The input of each layer is pooled by the compressive Haar transform for each layer: in the first layer input $X_1^{\mathrm{in}} = (x_{i,j}) \in \mathbb{R}^{8 \times d_1}$ is transformed by the compressive Haar basis matrix $\Phi_{8 \times 3}^{(0)}$ with size $8 \times 3$ formed by the first three column vectors, and the output is a feature array with size $3 \times d_1$; in the second layer $X_2^{\mathrm{in}} = (y_{i,j}) \in \mathbb{R}^{3 \times d_2}$ is transformed by the first column vector $\Phi_{3 \times 1}^{(1)}$ and the output is a feature vector with size $1 \times d_2$. In the plots of Haar basis matrix, the colors indicate the value of the entries of the Haar basis matrix.

synthesizes the node feature by adding the same weight to the nodes that are in the same cluster of the coarser layer, and in this way, pools the feature using the graph clustering information.

## 4  MATHEMATICS AND COMPUTATION FOR HAARPOOLING

**Chain of graphs by clustering**  For a graph $\mathcal{G} = (V, E, w)$, where $V, E, w$ are the vertices, edges, and weights on edges, a graph $\mathcal{G}^{\mathrm{cg}} := (V^{\mathrm{cg}}, E^{\mathrm{cg}}, w^{\mathrm{cg}})$ is a *coarse-grained graph* of $\mathcal{G}$ if $|V^{\mathrm{cg}}| \leq |V|$ and each node of $\mathcal{G}$ has only one parent node in $\mathcal{G}^{\mathrm{cg}}$ associated with it. Each node of $\mathcal{G}^{\mathrm{cg}}$ is called a *cluster* of $\mathcal{G}$. For integers $J > 0$, a *coarse-grained chain* for $\mathcal{G}$ is a sequence of graphs $\mathcal{G}_{0 \to J} := (\mathcal{G}_0, \mathcal{G}_1, \ldots, \mathcal{G}_J)$ with $\mathcal{G}_0 = \mathcal{G}$ and such that $\mathcal{G}_{j+1}$ is a coarse-grained graph of $\mathcal{G}_j = (V_j, E_j, w_j)$ for each $j = 0, 1, \ldots, J - 1$, and $\mathcal{G}_J$ has only one node. Here, we call the graph $\mathcal{G}_J$ the *top level* or the *coarsest level* and $\mathcal{G}_0$ the *bottom level* or the *finest level*. The chain $\mathcal{G}_{0 \to J}$ hierarchically coarsens graph $\mathcal{G}$. We use the notation $J + 1$ for the number of layers of the chain to distinguish the number $K$ of layers for pooling. The chain for graph $\mathcal{G}$ can be created by any clustering method. For details about graphs and chains, we refer the reader to the examples by Chung & Graham (1997); Hammond et al. (2011); Chui et al. (2015; 2018); Wang & Zhuang (2018; 2019).

### 4.1  COMPRESSIVE HAAR TRANSFORMS

**Haar basis**  The construction of Haar basis is rooted in the theory of Haar wavelet basis which was first introduced by Haar (1910). It is a special example of the more general Daubechies wavelets (Daubechies, 1992). Haar basis is later constructed on graph by Belkin et al. (2006), and also Chui et al. (2015); Wang & Zhuang (2018; 2019). The construction of the Haar basis is based on a chain of the graph. If the topology of the graph is well reflected by the clustering of the chain, then the Haar basis contains the crucial geometric information of the graph. For a chain $\mathcal{G}_{0 \to J}$, on the $j$th-layer graph $\mathcal{G}_j$, $j = 0, \ldots, J$, there is a Haar orthogonal basis $\{\phi_\ell^{(j)}\}_{\ell=0}^{N_j}$ defined on $\mathcal{G}_j$, where $N_j$ is the size of $\mathcal{G}_j$ and $N_{j+1} < N_j$ for $j = 0, \ldots, J - 1$. Suppose two consecutive

layers $j, j+1$. The first $N_{j+1}$ members of $\phi_\ell^{(j)}$, $\ell = 1, \ldots, N_{j+1}$, are defined on the finer layer $j+1$, and can be reduced into the $\phi_\ell^{(j+1)}$, $\ell = 1, \ldots, N_{j+1}$, as follows. For first $\ell = 1, \ldots, N_{j+1}$, $\phi_\ell^{(j)}(v) = \phi_\ell^{(j+1)}(Pa_{\mathcal{G}}(v))/\sqrt{|Pa_{\mathcal{G}}(v)|}$, i.e. the value of the $\phi_\ell^{(j)}(v)$ is equal to the scaled $\phi_\ell^{(j+1)}$ at the parent $Pa_{\mathcal{G}}(v)$ of $v$ and the scaled factor is one on square root of the number of nodes in the cluster which $v$ lies in. It means that $\phi_\ell^{(j)}(v)$ for $v$ sharing the parent have the same value. This property is critical to pooling as $\phi_\ell^{(j)}(v)$ can then be treated as weights for the graph $\mathcal{G}_j$ on which the input feature defined, and the nodes gain the same weight if they are in the same cluster. On the other hand, the remaining Haar basis vectors $\phi_\ell^{(j)}$ for $\ell = N_{j+1}+1, \ldots, N_j$ are constructed to reflect the high-frequency information in the Haar wavelet decomposition. This property is exploited by the compressive Haar basis which then pools the input feature into a lower (first) dimension output feature. The construction and its pseudo-codes for algorithmic implementation of the full Haar basis is detailed in Li et al. (2019); Wang & Zhuang (2019), which we also attach in the appendix. Let $\{\phi_\ell^{(j)}\}_{\ell=1}^{N_j}$, $j = 0, \ldots, J$, be the sequence of Haar bases associated with the layers of chain $\mathcal{G}_{0 \to J}$ of a graph $\mathcal{G}$. For $j = 0, \ldots, J$, we let the matrix $\widetilde{\Phi}_j = (\phi_1^{(j)}, \ldots, \phi_{N_j}^{(j)}) \in \mathbb{R}^{N_j \times N_j}$ and call the matrix $\widetilde{\Phi}_j$ *Haar transform matrix* for layer $j$.

**Orthogonality** For each level $j = 0, \ldots, J$, the sequence $\{\phi_\ell^{(j)}\}_{\ell=1}^{N_j}$, with $N_j := |V_j|$, is an orthonormal basis for the space $l_2(\mathcal{G}_j)$ of square-summable sequences on the graph $\mathcal{G}_j$, so that $(\phi_\ell^{(j)})^T \phi_{\ell'}^{(j)} = \delta_{\ell, \ell'}$. For each $j$, $\{\phi_\ell^{(j)}\}_{\ell=1}^{N_j}$ is the Haar basis system for the chain $\mathcal{G}_{j \to J}$.

**Locality** Let $\mathcal{G}_{0 \to J}$ be a coarse-grained chain for $\mathcal{G}$. If each parent of level $\mathcal{G}_j$, $j = 1, \ldots, J$, contains at least two children, the number of different scalar values of the components of a Haar basis vector $\phi_\ell^{(j)}$, $\ell = 1, \ldots, N_j$, is bounded by a constant independent of $j$.

In Figure 4, the Haar basis is created based on the coarse-grained chain $\mathcal{G}_{0 \to 2} := (\mathcal{G}_0, \mathcal{G}_1, \mathcal{G}_2)$, where $\mathcal{G}_0, \mathcal{G}_1, \mathcal{G}_2$ are graphs with $8, 3, 1$ nodes. The two colorful matrices show two Haar bases for the layers 0 and 1 in the chain $\mathcal{G}_{0 \to 2}$. There are in total 8 vectors of the Haar basis for $\mathcal{G}_0$ each with length 8, and 3 vectors of the Haar basis for $\mathcal{G}_1$ each with length 3. Haar basis matrix for each level of the chain has up to 3 different values in each column as indicated by colors in each matrix. For $j = 0, 1$, each node of $\mathcal{G}_j$ is a cluster of nodes in $\mathcal{G}_{j+1}$. Each column of the matrix is a member of the Haar basis on the individual layer of the chain. The first three column vectors of $\widetilde{\Phi}_1$ can be reduced as an orthonormal basis of $\mathcal{G}_1$ and the first column vector of $\mathcal{G}_1$ can be compressed to the constant basis for $\mathcal{G}_2$. This connection ensures that the compressive Haar transform for HaarPooling is feasible and would allow fast algorithms of HaarPooling (see Section 4.2 below).

**Adjoint and forward Haar transforms** We use adjoint Haar transforms for HaarPooling, which as the sparsity of the Haar basis matrix, the transform is fast implementable. The *adjoint Haar transform* for the signal $f$ on $\mathcal{G}_j$ is defined as

$$(\widetilde{\Phi}_j)^T f = \left( \sum_{v \in V} \phi_1^{(j)}(v) f(v), \ldots, \sum_{v \in V} \phi_{N_j}^{(j)}(v) f(v) \right) \in \mathbb{R}^{N_j}, \tag{3}$$

and the *forward Haar transform* for (coefficients) vector $c := (c_1, \ldots, c_{N_j}) \in \mathbb{R}^{N_j}$.

$$(\widetilde{\Phi}_j c)(v) = \sum_{\ell=1}^{N_j} \phi_\ell^{(j)}(v) c_\ell, \quad v \in V_j. \tag{4}$$

We call the components of $(\widetilde{\Phi}_j)^T f$ the *Haar (wavelet) coefficients* for $f$. The adjoint Haar transform represents the signal in Haar wavelet domain by computing the Haar coefficients for graph signal. Here the adjoint and forward Haar transforms can be extended to a feature data with size $N_j \times d_j$ by replacing the column vector $f$ by the feature array.

**Proposition 2.** *The adjoint and forward Haar Transforms are invertible in that for $j = 0, \ldots, J$ and vector $f$ on graph $\mathcal{G}_j$,*

$$f = \widetilde{\Phi}_j (\widetilde{\Phi}_j)^T f.$$

Proposition 2 shows that the forward Haar transform can recover the graph signal $f$ from the adjoint Haar transform $(\widetilde{\Phi}_j)^T f$. This means that adjoint and forward Haar transforms have zero-loss in graph signal transmission.

**Compressive Haar transforms**  Now for a graph neural network, suppose we want to use $K$ pooling layers. We associate the chain $\mathcal{G}_{0 \to K}$ of an input graph with the pooling by linking the $j$th layer of pooling with the $j$th layer of the chain. Then, we can use the Haar basis system on the chain to define the pooling operation. By the property of Haar basis, in the Haar transforms for layer $j$, $0 \le j \le K - 1$, of the $N_j$ Haar coefficients, the first $N_{j+1}$ coefficients are low-frequency coefficients, which reflect the approximation to the original data, and the other $(N_j - N_{j+1})$ coefficients are in high frequency, which contain fine details of the Haar wavelet decomposition. To define pooling, we remove the high-frequency coefficients in Haar wavelet representation and obtain the *compressive Haar transforms* for the feature $X_j^{\text{in}}$ at layers $j = 0, \ldots, K - 1$, which then gives the HaarPooling in Definition 1.

As shown in the following formula, the compressive Haar transform synthesizes the neighbourhood information of the signal $f$ as compared to the full Haar transform. Thus, HaarPooling takes the average information of the data $f$ over nodes in the same cluster.

$$\left\| \Phi_j^T X_j^{\text{in}} \right\|^2 = \sum_{p \in \mathcal{G}_{j+1}} \frac{1}{|Pa(v)|} \Big| \sum_{p=Pa(v)} X_j^{\text{in}}(v) \Big|^2, \quad \left\| \widetilde{\Phi}_j^T X_j^{\text{in}} \right\|^2 = \sum_{p \in \mathcal{G}_{j+1}} \sum_{p=Pa(v)} \Big| X_j^{\text{in}}(v) \Big|^2, \quad (5)$$

where $\widetilde{\Phi}_j$ is the full Haar basis matrix at the $j$th layer and $|Pa_{\mathcal{G}}(v)|$ means the number of nodes in the cluster which $v$ lies in. Here, in the first equation, $1/\sqrt{|Pa_{\mathcal{G}}(v)|}$ can be taken out of summation as $Pa(v)$ is in fact a set of nodes. We show the derivation of formula in equation 5 in Appendix D.

In HaarPooling, the compression or pooling occurs in the Haar wavelet domain. HaarPooling transforms the features on the nodes to the Haar wavelet domain and discards the high-frequency coefficients in the sparse Haar wavelet representation. Figure 4 shows a two-layer HaarPooling strategy. The first layer pools the input $X_0^{\text{in}}$ by the compressive Haar basis matrix $\Phi_{8 \times 3}^{(0)}$ to the output $X_0^{\text{out}}$ with lower first dimension. The second layer pools the input $X_1^{\text{in}}$ by the $\Phi_{3 \times 1}^{(1)}$ to the output $X_1^{\text{out}}$ which first dimension drops to one.

## 4.2   Fast Computation of HaarPooling

For the HaarPooling introduced in Definition 1, we can develop a fast computational strategy by virtue of fast adjoint Haar transforms. Let $\mathcal{G}_{0 \to K}$ be a coarse-grained chain of the graph $\mathcal{G}_0$. For convenience, we label the vertices of the level-$j$ graph $\mathcal{G}_j$ by $V_j := \big\{ v_1^{(j)}, \ldots, v_{N_j}^{(j)} \big\}$.

**Fast algorithm for HaarPooling**  The HaarPooling in equation 3 can be computed fast by using the hierarchical structure of the chain, as we introduce as follows. For $j = 1, \ldots, K$, let $c_k^{(j)}$ be the number of children of $v_k^{(j)}$, i.e. the number of vertices of $\mathcal{G}_{j-1}$ which belongs to the cluster $v_k^{(j)}$, for $k = 1, \ldots, N_j$. For $j = 0$, we let $c_k^{(0)} \equiv 1$ for $k = 1, \ldots, N_0$. Now, for $j = 0, \ldots, K$ and $k = 1, \ldots, N_j$, define the weight for the node $v_k^{(j)}$ of layer $j$ by

$$w_k^{(j)} := \frac{1}{\sqrt{c_k^{(j)}}}. \qquad (6)$$

Let $W_{0 \to K} := \{ w_k^{(j)} \mid j = 0, \ldots, K, \ k = 1, \ldots, N_j \}$. Then, for $j = 0, \ldots, K$, the weighted chain $(\mathcal{G}_{j \to K}, W_{j \to K})$ becomes a *filtration* if each parent of the chain $\mathcal{G}_{j \to K}$ has at least two children. See e.g. (Chui et al., 2015, Definition 2.3).

Let $j = 0, \ldots, K$. For the $j$th HaarPooling layer, let $\{ \phi_\ell^{(j)} \}_{\ell=1}^{N_j}$ be the Haar basis for the $j$th layer, which we also call the Haar basis for the filtration $(\mathcal{G}_{j \to K}, W_{j \to K})$ of a graph $\mathcal{G}$. For $k = 1, \ldots, N_j$, we let $X(v_k^{(j)}) = X(v_k^{(j)}, \cdot) \in \mathbb{R}^{d_j}$ the feature vector at node $v_k^{(j)}$. We define the weighted sum for feature $X \in \mathbb{R}^{N_j \times d_j}$ for $d_j \ge 1$ by

$$\mathcal{S}^{(j)}(X, v_k^{(j)}) := X(v_k^{(j)}), \quad v_k^{(j)} \in \mathcal{G}_j, \qquad (7)$$

---

**Algorithm 1:** Fast HaarPooling for One Layer

---

**Input:** Input feature $X_j^{\text{in}}$ for the $j$th pooling layer given $j = 0, \ldots, K-1$ in a GNN with total $K$ HaarPooling layers; the chain $\mathcal{G}_{j \to K}$ associated with the HaarPooling; numbers $N_i$ of nodes for layers $i = j, \ldots, K$.

**Output:** $\Phi_j^T X_j^{\text{in}}$ from Definition 1.

**Step 1:** Evaluate the sums for $i = j, \ldots, K$ recursively, using equation 7 and equation 8:
$$\mathcal{S}^{(i)}\big(X_j^{\text{in}}, v_k^{(i)}\big) \quad \forall v_k^{(i)} \in V_i .$$
**Step 2:**
**for** $\ell = 1$ **to** $N_{j+1}$ **do**

    Set $N_K = 0$.
    Compute $i$ such that $N_{i+1} + 1 \leq \ell \leq N_i$.
    Evaluate $\sum_{k=1}^{N_i} \mathcal{S}^{(i)}(X_j^{\text{in}}, v_k^{(i)}) w_k^{(i)} \phi_\ell^{(i)}(v_k^{(i)})$ in equation 9 by the two steps:
    (a) Compute the product for all $v_k^{(i)} \in V_i$:
      $T_\ell(X_j^{\text{in}}, v_k^{(i)}) = \mathcal{S}^{(i)}(X_j^{\text{in}}, v_k^{(i)}) w_k^{(i)} \phi_\ell^{(i)}(v_k^{(i)})$.
    (b) Evaluate sum $\sum_{k=1}^{N_i} T_\ell(X_j^{\text{in}}, v_k^{(i)})$.
**end for**

---

and recursively, for $i = j+1, \ldots, K$ and $v_k^{(i)} \in \mathcal{G}_i$,

$$\mathcal{S}^{(i)}\big(X, v_k^{(i)}\big) := \sum_{v_{k'}^{(i-1)} \in v_k^{(i)}} w_{k'}^{(i-1)} \mathcal{S}^{(i-1)}\big(X, v_{k'}^{(i-1)}\big). \tag{8}$$

For each vertex $v_k^{(i)}$ of $\mathcal{G}_i$, the $\mathcal{S}^{(i)}\big(X, v_k^{(i)}\big)$ is the weighted sum of the $\mathcal{S}^{(i-1)}\big(X, v_{k'}^{(i-1)}\big)$ at the level $i-1$ for those vertices $v_{k'}^{(i-1)}$ of $\mathcal{G}_{i-1}$ whose parent is $v_k^{(i)}$.

**Theorem 3.** *For $0 \leq j \leq K-1$, let $\{\phi_\ell^{(i)}\}_{\ell=1}^{N_i}$ for $i = j+1, \ldots, K$ be the Haar bases for the filtration $(\mathcal{G}_{j \to K}, W_{j \to K})$ at layer $i$. Then, the compressive Haar transform for the $j$th HaarPooling layer can be computed by, for the feature $X \in \mathbb{R}^{N_j \times d_j}$ and $\ell = 1, \ldots, N_j$,*

$$\big(\Phi_j^T X\big)_\ell = \sum_{k=1}^{N_i} \mathcal{S}^{(i)}\big(X, v_k^{(i)}\big) w_k^{(i)} \phi_\ell^{(i)}(v_k^{(i)}), \tag{9}$$

*where $i$ is the largest possible number in $\{j+1, \ldots, K\}$ such that $\phi_\ell^{(i)}$ is the $\ell$th member of the orthonormal basis $\{\phi_\ell^{(i)}\}_{\ell=1}^{N_i}$ for $l_2(\mathcal{G}_i)$, $v_k^{(i)}$ are the vertices of $\mathcal{G}_i$ and the weights $w_k^{(i)}$ are given by equation 12.*

We give the algorithmic implementation of Theorem 3 in Algorithm 1, which provides a fast algorithm for HaarPooling at each layer.

**Computational complexity**   With increasing graph size, the sparsity of the Haar basis matrix $\widetilde{\Phi}_j$ becomes more pronounced (Li et al., 2019). This sparsity implies fast computation for HaarPooling. The computational complexity of HaarPooling is determined by the adjoint Haar transforms. In the first step of Algorithm 1, the total number of summations for all elements of Step 1 is no more than $\sum_{i=0}^{j-1} N_{i+1}$; In the second step, by the locality of the Haar basis, the total number of multiplication and summation operations is at most $2 \sum_{\ell=1}^{N_j} C = \mathcal{O}(N_j)$. Here $C$ is the constant which bounds the number of different values of the Haar basis vector. Thus, the computational cost of Algorithm 1 is $\mathcal{O}(N_j)$.

We run an experiment to evaluate the CPU computational time of HaarPooling by Algorithm 1 against the direct matrix product. We use randomly generated graphs and features with size ranging from 2 to 5000. As shown in Figure 5, the fast HaarPooling has computational cost nearly proportional to the number of nodes $N$, while the ordinary matrix product incurs a cost close to order $\mathcal{O}(N^2)$. These results are consistent with the computational complexity analysis given above.

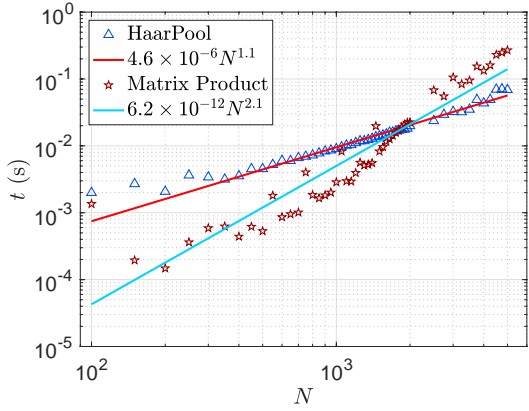

Figure 5: Comparison for fast computation and direct matrix product for HaarPooling for input feature array with up to 5000 nodes. The cost of HaarPooling has near linear computational complexity. The cost of direct matrix product grows at $\mathcal{O}(N^{2.1})$.

## 5 EXPERIMENTS

**Data sets** To verify whether the proposed framework can hierarchically learn good graph representations for classification, we evaluate *HaarPooling* on five widely used benchmark data sets for graph classification (Kersting et al., 2016), including one protein graph data set **PROTEINS** (Borgwardt et al., 2005; Dobson & Doig, 2003); two mutagen data sets **MUTAG** (Debnath et al., 1991; Kriege & Mutzel, 2012) and **MUTAGEN** (Riesen & Bunke, 2008; Kazius et al., 2005) (full name Mutagenicity); and two data sets that consist of chemical compounds screened for activity against non-small cell lung cancer and ovarian cancer cell lines, **NCI1** and **NCI109** (Wale et al., 2008). We include data sets from different domains, sample and graph sizes to give a comprehensive understanding of how *HaarPooling* performs with data sets in various scenarios. A summary information of the data sets is given in Table 1, which shows the data sets containing graphs with different sizes and structures: the number of data samples ranges from 188 to 4,337, the average number of nodes is from 17.93 to 39.06 and the average number of edges is from 19.79 to 72.82.

Table 1: Summary statistics of the data sets used in our experiments

| Data Set | MUTAG | PROTEINS | NCI1 | NCI109 | MUTAGEN |
|----------|-------|----------|------|--------|---------|
| max #nodes | 28 | 620 | 111 | 111 | 417 |
| min #nodes | 10 | 4 | 3 | 4 | 4 |
| avg #nodes | 17.93 | 39.06 | 29.87 | 29.68 | 30.32 |
| avg #edges | 19.79 | 72.82 | 32.30 | 32.13 | 30.77 |
| #graphs | 188 | 1,113 | 4,110 | 4,127 | 4,337 |
| #classes | 2 | 2 | 2 | 2 | 2 |

**Baselines and running environment** We compare **HaarPool** with **SortPool** (Zhang et al., 2018a), **DiffPool** (Ying et al., 2018), **gPool** (Gao & Ji, 2019), **SAGPool** (Lee et al., 2019), **EigenPool** (Ma et al., 2019a), **CSM** (Kriege & Mutzel, 2012) and **GIN** (Xu et al., 2019) on the above data sets. The experiments use PyTorch Geometric[1] (Fey & Lenssen, 2019) and were run in Google Cloud using 4 Nvidia Telsa T4 with 2560 CUDA cores, compute 7.5, 16GB GDDR6 VRAM.

**Training procedures** In experiments, we use a GNN with at most 3 GCN (Kipf & Welling, 2017) convolutional layers plus one HaarPooling layer, followed by three fully connected layers. The hyperparameters of the network are adjusted case by case. We use spectral clustering, which exploits the eigenvalues of the graph Laplacian, to generate a chain with the number of layers given. Spectral clustering has shown good performance in coarsening a variety of data patterns and can handle isolated nodes.

---

[1]`https://pytorch-geometric.readthedocs.io/en/latest.`

We use random shuffling of the data set, which we split into training, validation, and test sets with proportions 80%, 10% and 10% respectively. We use the Adam optimizer (Kingma & Ba, 2015), early stopping criterion, patience. The specific values are provided in the appendix. The early stopping criterion was that the validation loss does not improve for 50 epochs, with a maximum of 150 epochs, as suggested by Shchur et al. (2018).

**Results**  The classification test accuracy is reported in Table 6. GNNs with HaarPooling have excellent performance on all data sets. In 4 out of 5 datasets, it achieved top accuracy. This shows that HaarPooling with appropriate graph convolution, can achieve top performance on a variety of graph classification tasks, and in some cases improve state of the art by a few percent points.

Table 2: Performance comparison for graph classification tasks (test accuracy in percent, showing the standard deviation over 10 repetitions of the experiment).

| Method | MUTAG | PROTEINS | NCI1 | NCI109 | MUTAGEN |
|---|---|---|---|---|---|
| CSM | 85.4 | – | – | – | – |
| GIN | 89.4 | 76.2 | **82.7** | – | – |
| SortPool | 85.8 | 75.5 | 74.4 | 72.3* | 78.8* |
| DiffPool | – | 76.3 | 76.0* | 74.1* | 80.6* |
| gPool | – | 77.7 | – | – | – |
| SAGPool | – | 72.1 | 74.2 | 74.1 | – |
| EigenPool | – | 76.6 | 77.0 | 74.9 | 79.5 |
| HaarPool | **90.0±3.6** | **80.4±1.8** | 78.6±0.5 | **75.6±1.2** | **80.9±1.5** |

'*' means that the records are retrieved from EigenPool (Ma et al., 2019a), '–' means that there is no public records for the corresponding method on the data set, and the bold number indicates the best performance in the list.

## 6 CONCLUSION

We introduced a new graph pooling method called HaarPooling. HaarPooling has a mathematical formalism derived from compressive Haar transforms. Unlike existing graph pooling methods, HaarPooling takes into account both the graph structure and also the features over the nodes of the graph-structured input data, to compute a coarsened representation. As an individual unit, HaarPooling can be applied in conjunction with any type of graph convolution in GNNs. We show in experiments that HaarPooling reaches state of the art in several benchmark graph classification tasks. Moreover, having only linear computational complexity in the size of the inputs, HaarPooling is a very fast pooling method.

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

## A  GRAPH CLASSIFICATION

**Graph classification**  This task is to categorize graph-structured data into several classes. The training set consists of $M$ pairs of samples $\big((x_i, \mathcal{G}_i), y_i\big)$, $i = 1, \ldots, M$. For the $i$th sample, $\mathcal{G}_i = (V_i, E_i, W_i)$ is a graph with vertex set $V_i$ of size $|V_i| = N_i$ (also called nodes), and edge set $E_i$ with weights $W_i$. The feature $x_i \in \mathbb{R}^{N_i \times d}$ is an array of $d$ features per vertex, i.e., an $\mathbb{R}^d$-valued function over $V_i$. The label $y_i$ is an integer from a finite set indicating which class the input sample $(x_i, \mathcal{G}_i)$ lies in. The number of nodes $N_i$ and the graph structure $E_i, W_i$ usually vary over the different input samples.

**Graph neural networks**  Deep graph neural networks (GNNs) are designed to work with graph-structured inputs of the form $(x_i, \mathcal{G}_i)$ described above. A GNN is typically composed of multiple *graph convolution* layers, *graph pooling* layers, and fully connected layers. A (graph) convolutional layer extracts an array of features from the previous array. It changes the dimension $d$ of the feature array but does not change the number of nodes $N_i$. Since the number of nodes of different inputs is variable, the number of nodes of the corresponding outputs is also variable. This raises new challenges in comparison with traditional image classification tasks, where the local structure connecting pixels is always fixed (even if the number of pixels might be variable).

**Graph pooling**  In GNNs, one uses graph pooling to reduce the first dimension $N$ of the feature arrays, and more importantly, to obtain outputs of uniform dimension (commonly followed by fully connected layers). A general architecture uses a cascade of convolutional and pooling layers. Figure 2 illustrates such an architecture with three blocks of graph convolutional and pooling layers, followed by a multi-layer perceptron (MLP) with three fully connected layers. In practice, each block can include several convolutional layers but use only one pooling layer at most. The exact architecture of GNNs with combined convolutional and pooling layers is mainly dependent upon the particular problem and the data set and is designed case by case.

## B  CONSTRUCTION OF HAAR BASIS

**Construction of Haar basis.**  With a chain of the graph, one can generate a Haar basis for $l_2(\mathcal{G})$ following Chui et al. (2015), see also Gavish et al. (2010). We show the construction of Haar basis on $\mathcal{G}$, as follows.

**Step 1.** Let $\mathcal{G}^{\mathrm{cg}} = (V^{\mathrm{cg}}, E^{\mathrm{cg}}, w^{\mathrm{cg}})$ be a coarse-grained graph of $\mathcal{G} = (V, E, w)$ with $N^{\mathrm{cg}} := |V^{\mathrm{cg}}|$. Each vertex $v^{\mathrm{cg}} \in V^{\mathrm{cg}}$ is a cluster $v^{\mathrm{cg}} = \{v \in V \mid v \text{ has parent } v^{\mathrm{cg}}\}$ of $\mathcal{G}$. Order $V^{\mathrm{cg}}$, e.g., by degrees of vertices or weights of vertices, as $V^{\mathrm{cg}} = \{v_1^{\mathrm{cg}}, \ldots, v_{N^{\mathrm{cg}}}^{\mathrm{cg}}\}$. We define $N^{\mathrm{cg}}$ vectors $\phi_\ell^{\mathrm{cg}}$ on $\mathcal{G}^{\mathrm{cg}}$ by

$$\phi_1^{\mathrm{cg}}(v^{\mathrm{cg}}) := \frac{1}{\sqrt{N^{\mathrm{cg}}}}, \quad v^{\mathrm{cg}} \in V^{\mathrm{cg}}, \tag{10}$$

and for $\ell = 2, \ldots, N^{\mathrm{cg}}$,

$$\phi_\ell^{\mathrm{cg}} := \sqrt{\frac{N^{\mathrm{cg}} - \ell + 1}{N^{\mathrm{cg}} - \ell + 2}} \left( \chi_{\ell-1}^{\mathrm{cg}} - \frac{\sum_{j=\ell}^{N^{\mathrm{cg}}} \chi_j^{\mathrm{cg}}}{N^{\mathrm{cg}} - \ell + 1} \right), \tag{11}$$

where $\chi_j^{\mathrm{cg}}$ is the indicator function for the $j$th vertex $v_j^{\mathrm{cg}} \in V^{\mathrm{cg}}$ on $\mathcal{G}$ given by

$$\chi_j^{\mathrm{cg}}(v^{\mathrm{cg}}) := \begin{cases} 1, & v^{\mathrm{cg}} = v_j^{\mathrm{cg}}, \\ 0, & v^{\mathrm{cg}} \in V^{\mathrm{cg}} \backslash \{v_j^{\mathrm{cg}}\}. \end{cases}$$

Then, one can show that $\{\phi_\ell^{\mathrm{cg}}\}_{\ell=1}^{N^{\mathrm{cg}}}$ forms an orthonormal basis for $l_2(\mathcal{G}^{\mathrm{cg}})$.

Note that each $v \in V$ belongs to exactly one cluster $v^{\mathrm{cg}} \in V^{\mathrm{cg}}$. In view of this, for each $\ell = 1, \ldots, N^{\mathrm{cg}}$, we can extend the vector $\phi_\ell^{\mathrm{cg}}$ on $\mathcal{G}^{\mathrm{cg}}$ to a vector $\phi_{\ell,1}$ on $\mathcal{G}$ by

$$\phi_{\ell,1}(v) := \frac{\phi_\ell^{\mathrm{cg}}(v^{\mathrm{cg}})}{\sqrt{|v^{\mathrm{cg}}|}}, \quad v \in v^{\mathrm{cg}},$$

here $|v^{\mathrm{cg}}| := k_\ell$ is the size of the cluster $v^{\mathrm{cg}}$, i.e., the number of vertices in $\mathcal{G}$ whose common parent is $v^{\mathrm{cg}}$. We order the cluster $v_\ell^{\mathrm{cg}}$, e.g., by degrees of vertices, as

$$v_\ell^{\mathrm{cg}} = \{v_{\ell,1}, \ldots, v_{\ell,k_\ell}\} \subseteq V.$$

For $k = 2, \ldots, k_\ell$, similar to equation 11, define

$$\phi_{\ell,k} = \sqrt{\frac{k_\ell - k + 1}{k_\ell - k + 2}} \left( \chi_{\ell,k-1} - \frac{\sum_{j=k}^{k_\ell} \chi_{\ell,j}}{k_\ell - k + 1} \right).$$

where for $j = 1, \ldots, k_\ell$, $\chi_{\ell,j}$ is given by

$$\chi_{\ell,j}(v) := \begin{cases} 1, & v = v_{\ell,j}, \\ 0, & v \in V \backslash \{v_{\ell,j}\}. \end{cases}$$

One can show that the resulting $\{\phi_{\ell,k} : \ell = 1, \ldots, N^{\mathrm{cg}}, k = 1, \ldots, k_\ell\}$ is an orthonormal basis for $l_2(\mathcal{G})$.

**Step 2.** Let $\mathcal{G}_{0 \to J}$ be a coarse-grained chain for the graph $\mathcal{G}$. An orthonormal basis $\{\phi_\ell^{(J)}\}_{\ell=1}^{N_J}$ for $l_2(\mathcal{G}_J)$ is generated using equation 10 and equation 11. We then repeatedly use Step 1: for $j = 0, \ldots, J-1$, we generate an orthonormal basis $\{\phi_\ell^{(j)}\}_{\ell=1}^{N_j}$ for $l_2(\mathcal{G}_j)$ from the orthonormal basis $\{\phi_\ell^{(j+1)}\}_{\ell=1}^{N_{j+1}}$ for the coarse-grained graph $\mathcal{G}_{j+1}$ that was derived in the previous steps. We call the sequence $\{\phi_\ell := \phi_\ell^{(0)}\}_{\ell=1}^{N_0}$ of vectors at the finest level, the *Haar global orthonormal basis* or simply the *Haar basis* for $\mathcal{G}$ associated with the chain $\mathcal{G}_{0 \to J}$. The orthonormal basis $\{\phi_\ell^{(j)}\}_{\ell=1}^{N_j}$ for $l_2(\mathcal{G}_j)$, $j = 1, \ldots, J$ is called the *associated (orthonormal) basis* for the Haar basis $\{\phi_\ell\}_{\ell=1}^{N}$.

Besides the orthogonality, the Haar basis has the locality which is critical to the fast computation of HaarPooling.

**Compressive Haar basis** Suppose we have constructed the (full) Haar basis $\{\phi_\ell^{(j)}\}_{\ell=0}^{N_j}$ for each layer $\mathcal{G}_j$ of the chain $\mathcal{G}_{0 \to K}$. The *compressive Haar basis* for layer $j$ is $\{\phi_\ell^{(j)}\}_{\ell=0}^{N_{j+1}}$.

## C   FAST COMPUTATION FOR HAARPOOLING

Let $\mathcal{G}_{0 \to K}$ be a coarse-grained chain of the graph $\mathcal{G}_0$. For convenience, we label the vertices of the level-$j$ graph $\mathcal{G}_j$ by $V_j := \{v_1^{(j)}, \ldots, v_{N_j}^{(j)}\}$.

**Fast algorithm for HaarPooling** The HaarPooling in equation 3 can be computed fast by using the hierarchical structure of the chain, as we introduce as follows. For $j = 1, \ldots, K$, let $c_k^{(j)}$ be the number of children of $v_k^{(j)}$, i.e. the number of vertices of $\mathcal{G}_{j-1}$ which belongs to the cluster $v_k^{(j)}$, for $k = 1, \ldots, N_j$. For $j = 0$, we let $c_k^{(0)} \equiv 1$ for $k = 1, \ldots, N_0$. Now, for $j = 0, \ldots, K$ and $k = 1, \ldots, N_j$, define the weight for the node $v_k^{(j)}$ of layer $j$ by

$$w_k^{(j)} := \frac{1}{\sqrt{c_k^{(j)}}}. \tag{12}$$

Let $W_{0 \to K} := \{w_k^{(j)} \mid j = 0, \ldots, K, \ k = 1, \ldots, N_j\}$. Then, for $j = 0, \ldots, K$, the weighted chain $(\mathcal{G}_{j \to K}, W_{j \to K})$ becomes a *filtration* if each parent of the chain $\mathcal{G}_{j \to K}$ has at least two children. See e.g. (Chui et al., 2015, Definition 2.3).

## D  PROOFS

*Proof for equation 5.* We only need to prove the first formula. The second is obtained by definition. To simplify notation, we let $f = X_j^{\mathrm{in}}$. By construction of Haar basis, for some layer $j$, the first $N_{j+1}$ basis vectors

$$\phi_\ell^{(j)}(v) = \phi_\ell^{(j+1)}(p)/\sqrt{|Pa_\mathcal{G}(v)|}, \quad \text{for } p = Pa_\mathcal{G}(v).$$

Then, the Fourier coefficient of $f$ for the $\ell$th basis vector is the inner product

$$
\begin{aligned}
\left\langle f, \phi_\ell^{(j)} \right\rangle &= \sum_{v \in \mathcal{G}_j} f(v)\overline{\phi_\ell^{(j)}(v)} \\
&= \sum_{p \in \mathcal{G}_{j+1}} \sum_{p = Pa_\mathcal{G}(v)} f(v)\overline{\phi_\ell^{(j+1)}(p)}/\sqrt{|Pa_\mathcal{G}(v)|} \\
&= \sum_{p \in \mathcal{G}_{j+1}} \widetilde{f}(p)\overline{\phi_\ell^{(j+1)}(p)} \\
&= \left\langle \widetilde{f}, \phi_\ell^{(j+1)} \right\rangle
\end{aligned}
$$

where we have let

$$\widetilde{f}(p) := \frac{1}{\sqrt{|Pa_\mathcal{G}(v)|}} \sum_{p = Pa_\mathcal{G}(v)} f(v).$$

This then gives

$$\sum_{\ell=1}^{N_{j+1}} \left| \left\langle f, \phi_\ell^{(j)} \right\rangle \right|^2 = \sum_{\ell=1}^{N_{j+1}} \left| \left\langle \widetilde{f}, \phi_\ell^{(j+1)} \right\rangle \right|^2. \tag{13}$$

Since $\{\phi_\ell\}_{\ell=1}^{N_{j+1}}$ forms an orthonormal basis on $\ell_2(\mathcal{G}_{j+1})$,

$$
\left\| \Phi_j^T f \right\|^2 = \sum_{\ell=1}^{N_{j+1}} \left| \left\langle \widetilde{f}, \phi_\ell^{(j+1)} \right\rangle \right|^2 = \left\| \widetilde{f} \right\|^2 = \sum_{p \in \mathcal{G}_{j+1}} \left| \widetilde{f}(p) \right|^2
$$

$$
= \sum_{p \in \mathcal{G}_{j+1}} \left| \frac{1}{\sqrt{|Pa_\mathcal{G}(v)|}} \sum_{p = Pa_\mathcal{G}(v)} f(v) \right|^2.
$$

This proves the left formula in equation 5. $\qquad\square$

*Proof of Theorem 3.* By the relation between $\phi_\ell^{(i)}$ and $\phi_\ell^{(j)}$, for $i = j+1, \ldots, K$ and $\ell = 1, \ldots, N_{j+1}$,

$$
\begin{aligned}
\left(\Phi_j^T X\right)_\ell &= \sum_{k=1}^{N_j} X(v_k^{(j)}) \phi_\ell^{(j)}(v_k^{(j)}) \\
&= \sum_{k'=1}^{N_{j+1}} \left( \sum_{Pa_\mathcal{G}(v_k^{(j)})=v_{k'}^{(j+1)}} X(v_k^{(j)}) \right) w_{k'}^{(j+1)} \phi_\ell^{(j+1)}(v_{k'}^{(j+1)}) \\
&= \sum_{k'=1}^{N_{j+1}} \mathcal{S}^{(j+1)}(X, v_{k'}^{(j+1)}) w_{k'}^{(j+1)} \phi_\ell^{(j+1)}(v_{k'}^{(j+1)}) \\
&= \sum_{k''=1}^{N_{j+2}} \left( \sum_{Pa_\mathcal{G}(v_{k'}^{(j+1)})=v_{k''}^{(j+2)}} \mathcal{S}^{(j+1)}(X, v_{k'}^{(j+1)}) w_{k'}^{(j+1)} \right) w_{k''}^{(j+2)} \phi_\ell^{(j+2)}(v_{k''}^{(j+2)}) \\
&= \sum_{k''=1}^{N_{j+2}} \mathcal{S}^{(j+2)}(X, v_{k''}^{(j+2)}) w_{k''}^{(j+2)} \phi_\ell^{(j+2)}(v_{k''}^{(j+2)}) \\
&\quad \ldots\ldots \\
&= \sum_{k=1}^{N_i} \mathcal{S}^{(i)}(X, v_k^{(i)}) w_k^{(i)} \phi_\ell^{(i)}(v_k^{(i)}),
\end{aligned}
$$

where $v_{k'}^{(j+1)}$ is the parent of $v_k^{(j)}$ and $v_{k''}^{(j+2)}$ is the parent of $v_k^{(j+1)}$, and we recursively compute the summation to obtain the last equality, thus completing the proof. $\square$

## E  EXPERIMENTAL SETTING

The architecture of GNN is identified by the layer type and the number of hidden nodes at each layer. For example, we denote 3GC256-HP-2FC256-FC128 to represent a GNN architecture with 3 GCNConv layers each with 256 hidden nodes plus one HaarPooling layer followed by 2 fully connected layers each with 256 hidden nodes and 1 fully connected layer with 128 hidden nodes. The architecture for each data set is shown by Table 3.

The hyperparameters include batch size; learning rate, weight decay rate (these two for optimization); maximum number of epochs; patience for early stopping. The choice of hyperparameters in each data set is shown in Table 4.

Table 3: Network architecture

| Data Set | Layers and #Hidden Nodes |
|---|---|
| MUTAG | GC60-HP-FC60-FC180-FC60 |
| PROTEINS | 2GC128-HP-2GC128-HP-2GC128-HP-GC128-2FC128-FC64 |
| NCI1 | 2GC256-HP-FC256-FC1024-FC2048 |
| NCI109 | 3GC256-HP-2FC256-FC128 |
| MUTAGEN | 3GC256-HP-2FC256-FC128 |

## F  GNN WITH HAARPOOLING ON TRIANGLES

We test GNN with HaarPooling on graph data set **Triangles** (Knyazev et al., 2019). **Triangles** is a 10 classification problem with 45000 graphs. The average numbers of nodes and edges of graphs are

Table 4: Hyperparameter setting

| Data Set | MUTAG | PROTEINS | NCI1 | NCI109 | MUTAGEN |
|---|---|---|---|---|---|
| batch size | 60 | 50 | 100 | 100 | 100 |
| max #epochs | 30 | 20 | 150 | 150 | 50 |
| early stopping | 15 | 20 | 50 | 50 | 50 |
| learning rate | 0.01 | 0.001 | 0.001 | 0.01 | 0.01 |
| weight decay | 0.0005 | 0.0005 | 0.0005 | 0.0001 | 0.0005 |

20.85 and 32.74 respectively. In the experiments, the network uses GIN convolution (Xu et al., 2019) as graph convolution and with HaarPooling or SAGPooling (Lee et al., 2019). With SAGPooling, the network architecture uses two combined layers of GIN convolution and SAGPooling followed by combined layers of GIN convolution and *global max pooling*, denoted by GIN-SP-GIN-SP-GIN-MP, where SP means the SAGPooling and MP means global max pooling. With HaarPooling, we test with two architectures: GIN-HP-GIN-HP-GIN-MP and GIN-HP-GIN-GIN-MP, where HP means HaarPooling. The data for training, validation and test are 35000, 5000 and 10000 respectively. The hidden nodes in convoluational layers is 64, batch size is 60 and learning rate is 0.001.

Table 5 shows the test accuracy of the three networks. It illustrates that both networks with Haar-Pooling outperform that with the SAGPooling.

Table 5: Training, validation and test accuracies on Triangles

| Architecture | Accuracy (%) | | |
|---|---|---|---|
| | Training | Validation | Test |
| GIN-SP-GIN-SP-GIN-MP | 45.6 | 45.3 | 44.0 |
| GIN-HP-GIN-HP-GIN-MP | 47.5 | 46.3 | 46.1 |
| GIN-HP-GIN-GIN-MP | 47.3 | 45.8 | 45.5 |

## G PROPERTY COMPARISON OF POOLING METHODS

Here we provide a comparison of the properties of HaarPooling with existing pooling methods. The properties in comparison includes time complexity and space complexity, and whether involving the clustering, hierarchical pooling (which is then not a global pooling), spectral-based, node feature or graph structure and sparse representation. We compare HaarPooling (denoted by HaarPool in the table) to other methods (SortPool, DiffPool, gPool, SAGPool and EigenPool). The SortPool (i.e. Sort-Pooling) is a global pooling which uses node signature (i.e. Weisfeiler-Lehman color of vertex) sorts all vertices by the values of the channels of the input data. Thus, the time complexity (worst case) of SortPool is $\mathcal{O}(|V|^2)$ and space complexity is $\mathcal{O}(|V|)$. Other pooling methods are all hierarchical pooling. DiffPool and gPool both use the node feature and have time complexity $\mathcal{O}(|V|^2)$ The Diff-Pool learns the assignment matrices in end-to-end manner and has space complexity $\mathcal{O}(k|V|^2)$ for pooling ratio $k$. The gPool projects all nodes to a learnable vector to generate scores for nodes, and then sorts the nodes by the projection scores; the space complexity is $\mathcal{O}(|V| + |E|)$. SAGPool uses the graph convolution to calculate the attention scores of nodes and then selects top ranked nodes for pooling. The time complexity of SAGPool is $\mathcal{O}(|E|)$ and the space complexity is $\mathcal{O}(|V| + |E|)$ due to the sparsity of the pooling matrix. EigenPool, which considers both the node feature and graph structure, uses the eigedecomposition of subgraphs (from clustering) of the input graph, and pools the input data by Fourier transforms of the assembled basis matrix. Due to eigendecomposition, the time complexity of EigenPool is $\mathcal{O}(|V|^2)$ and space complexity is $\mathcal{O}(|V|^2)$. HaarPool which uses the sparse representation of data by compressive Haar basis has linear time complexity $\mathcal{O}(|V|)$ (up to a $\log |V|$ term), and the space complexity is $\mathcal{O}(|V|^2 \epsilon)$, where $\epsilon$ is the sparsity of the compressive Haar transform matrix and is usually very small. From the table, we can observe the HaarPool is the

Table 6: Property comparison for pooling methods.

| Method | Time Complexity | Space Complexity | Clustering-based | Spectral-based | Hierarchical Pooling | Use Node Feature | Use Graph Structure | Sparse Representation |
|--------|-----------------|------------------|------------------|----------------|----------------------|------------------|---------------------|----------------------|
| SortPool | $\mathcal{O}(|V|^2)$ | $\mathcal{O}(|V|)$ | | | | ✓ | | |
| DiffPool | $\mathcal{O}(|V|^2)$ | $\mathcal{O}(k|V|^2)$ | | | ✓ | ✓ | | |
| gPool | $\mathcal{O}(|V|^2)$ | $\mathcal{O}(|V|+|E|)$ | | | ✓ | ✓ | | |
| SAGPool | $\mathcal{O}(|E|)$ | $\mathcal{O}(|V|+|E|)$ | | | ✓ | ✓ | ✓ | |
| EigenPool | $\mathcal{O}(|V|^2)$ | $\mathcal{O}(|V|^2)$ | ✓ | ✓ | ✓ | ✓ | ✓ | |
| HaarPool | $\mathcal{O}(|V|)$ | $\mathcal{O}(|V|^2\epsilon)$ | ✓ | ✓ | ✓ | ✓ | ✓ | ✓ |

'$|V|$' is the number of vertices of the input graph; '$|E|$' is the number of edges of the input graph; '$\epsilon$' in HaarPooling is the sparsity of the compressive Haar transform matrix; '$k$' in the DiffPool is the pooling ratio.

only pooling method which has time complexity proportional to the number of nodes, and thus has faster implementation.

