# OpenReview forum: "HaarPooling: Graph Pooling with Compressive Haar Basis"
_ICLR.cc/2020/Conference — Reject_

### Official Review · AnonReviewer1 · 2019-10-23
**Official Blind Review #1**

**Rating:** 3

**Review:**

Summary

This paper proposed a new graph pooling method based on the Haar basis on graphs. The authors argued that existing graph pooling methods have drawbacks (ignored node features, ignored the hierarchical structure of a graph or, computationally expensive in terms of space or time) and that the proposed method overcame these drawbacks. The proposed method extracts the low-frequency components of signals in terms of Haar decomposition on a chain of graphs. By the property of the Haar basis, we can compute the pooling matrix with the complexity proportional to the graph size. The paper empirically compared the proposed method with state-of-the-art pooling methods and graph NNs using well-known datasets.


Decision

Although this paper gave a novel pooling method by incorporating the Wavelet theory, I still have questions for the effectiveness of the proposed method in the real datasets and whether the proposed method solved the problems of existing methods the authors mentioned in the introduction (see the Suggestion section). Besides, I think the authors can improve the organization of the paper to maximize the value of the paper. Therefore, I judge the paper as a border, tending to reject for now.


Suggestions

- The main part of the paper is 10 pages long. I think the authors can polish the organization of the paper to fit the recommended page size (i.e., 8 pages).
- I want to know how many times the authors ran experiments for each configuration.  I think the variance of test accuracies are critical since the difference in performance between methods is not significant. Adding error bars to the experiment results are preferrable if authors ran experiments multiple times.
- The authors claimed in the introduction that the drawback of existing methods is their time or space complexity and that the proposed method is computationally inexpensive. I think it is better to emphasize that HaarPooling does solve the questions. Although the authors demonstrated that the proposed algorithm is fast compared to the naive implementation, the comparison with other pooling methods are missing. For example, adding a comparison table in terms of time and space complexity of pooling methods is one idea. Another idea is to demonstrate it empirically by providing time and memory consumption to the experiment results.


Minor Comments

- Introduction
	- Haar Pooling is computed following a chain... → Haar Pooling is computed by following ...
- page 1, section 1, paragraph 1
	- Graph classification and regression are a very different kind of task
	- → At first sight, I have thought that this sentence discusses the difference between classification and regression tasks. Could you reconsider the sentence?
- page 3, section 2, paragraph 1
	- The authors used the term "chain," whose definition is available in the later section (Sec. 3). I think this terminology is not common in the literature of graph NNs (at least I did not come up with the definition from this term). Could you consider to add the definition (or brief explanation) of the term when you use it for the first time?
- page 5, section 3, Chain of graph by clustering
	- Write what $w$ is in the definition of a graph $\mathcal{G}=(V, E, w)$.
- page 5, section 3.1, Orthogonality
	- Define $l_2(\mathcal{G}_j)$ (I can imagine its definition, though).
- page 5, section 3.1, Haar basis
	- For two consective layers $j$, $j+1$ ...
	- → I had a little difficulty in understanding this sentence. Could you reconsider the wording?
-  page 13, Appendix B, (10)
	-  I think "1" in this equation is an all-one vector. It is better to write differently the all-one vector and the scalar one.
-  page 5, section 3.1 and page 7, (8)
	-  For the node $v$ in the $j$-th graph and $v'$ in the $(j+1)$-th graph, the authors use both of $v\in v'$ (section 3.1) and $v'\in Pa_{\mathcal{G}}(v)$ (equation (8)) to denote the parent-child relationship. It is better to align the notation.
- page 13, Appendix B
	- Could you write in the appendix how to construct the compressed Haar from the full counterpart (I suppose we should choose bases corresponding to $k=1$)?


Questions

- In the paragraph starting with Locality on page 5, the paper says that the number of different values of the Haar basis is bounded by constant. I want the authors to make this sentence more precise. To be more specific, what "the different values" means (I imagine that it means the number of scalar values appearing as a component of some basis) and what "constant" means (with respect to which variable the "constant" is constant?)
- I think that the complexity of the direct matrix product is $O(N^2C)$, where $N$ is the graph size and $C$ is the channel size (correct me if I am wrong). Therefore the result of Figure 5 was surprising to me because the direct matrix product takes approximately $O(N^3)$ time.  These look inconsistent to me unless the channel size $C$ is proportional to the node size $N$.
- Equation (5) claimed that the compressive Haar basis approximately keeps the length of an input signal. I want to know how we can justify it (theoretically or empirically).  I think that it is empirically correct for the Fourier basis that signal carries "information" in the low-frequency domains and that high-frequency areas are noisy (e.g., NT and Maehara (2019); Ma et al., 2019). If this is the case for Haar transform, too, I understand that equation (5) holds and gives the justification that the proposed method can extract the information from lower frequencies.

**Experience Assessment:**

I have published one or two papers in this area.

**Review Assessment: Checking Correctness Of Derivations And Theory:**

I assessed the sensibility of the derivations and theory.

**Review Assessment: Checking Correctness Of Experiments:**

I assessed the sensibility of the experiments.

**Review Assessment: Thoroughness In Paper Reading:**

I read the paper at least twice and used my best judgement in assessing the paper.

---

> ### Author Response · Authors · 2019-11-13
> **Responses to Reviewer #1 (Part 1 of 2)**
>
> Decision
>
> 1. Although this paper gave a novel pooling method by incorporating the Wavelet theory, I still have questions for the
> effectiveness of the proposed method in the real datasets and whether the proposed method solved the problems
> of existing methods the authors mentioned in the introduction (see the Suggestion section).Besides, I think the
> authors can improve the organization of the paper to maximize the value of the paper.
>
> Thank you for the helpful comments. We have provided our responses below and changed the paper with your suggestion.
>
> Suggestion
>
> 2. The main part of the paper is 10 pages long. I think the authors can polish the organization of the paper to fit the
> recommended page size (i.e., 8 pages).
>
> We have re-organized the paper and the modified version has around 9 pages.
>
> 3. I want to know how many times the authors ran experiments for each configuration. I think the variance of test
> accuracies are critical since the difference in performance between methods is not significant. Adding error bars to the experiment results are preferrable if authors ran experiments multiple times.
>
> We had repeatedly run each experiment 10 times where the dataset uses random shuffle. In the revision, we added the standard deviation for each test accuracy which was not mentioned previously. HaarPooling has the top test accuracy with small variance for four data sets (MUTAG, PROTEINS, NCI109 and MUTAGEN). The conclusion thus remains as previously reported.
>
> 4. The authors claimed in the introduction that the drawback of existing methods is their time or space complexity
> and that the proposed method is computationally inexpensive. I think it is better to emphasize that HaarPooling
> does solve the questions. Although the authors demonstrated that the proposed algorithm is fast compared to the
> naive implementation, the comparison with other pooling methods are missing. For example, adding a comparison
> table in terms of time and space complexity of pooling methods is one idea. Another idea is to demonstrate it
> empirically by providing time and memory consumption to the experiment results.
>
> We provided Table 5 in Appendix G for comparison of time and space computational complexity with other pooling methods.
> Our method has linear time complexity of number of nodes which outperforms the other methods.
> Besides, we also compare the properties of the algorithm on whether involving the clustering, hierarchical pooling (which is then not a global pooling), spectral-based, node feature or graph structure and sparse representation.
>
> 5. Haar Pooling is computed following a chain... $\to $ Haar Pooling is computed by following ...
>
> Improved this.
>
> 6. page 1, section 1, paragraph 1: Graph classification and regression are a very different kind of task
> $\to$ At first sight, I have thought that this sentence discusses the difference between classification and
> regression tasks. Could you reconsider the sentence?
>
> Yes. We changed it to 'Differe from node classification,  graph classification is a task where the label of any given graph-structured sample is to be predicted based on a training set of labeled graph-structured samples.'.
>
> 7. page 3, section 2, paragraph 1: The authors used the term "chain," whose definition is available in the later section (Sec. 3). I think this terminology is not common in the literature of graph NNs (at least I did not come up with the definition from this term). Could you consider to add the definition (or brief explanation) of the term when you use it for the first time?
>
> Yes. We now include a short description 'i.e., a sequence of graphs $(\mathcal{G}_0, \mathcal{G}_1, \ldots, \mathcal{G}_K)$, where the nodes of each $\mathcal{G}_{j+1}$ correspond to clusters of nodes of $\mathcal{G}_{j+1}$, $j=0,\ldots,K-1$'.
>
> 8. page 5, section 3, Chain of graph by clustering: Write what is $w$ in the definition of a graph $\mathcal{G}=(V,E,w)$.
>
> To explain that term, we modified the first sentence as 'For a graph $\mathcal{G}=(V,E,w)$, where $V,E,w$ are the vertices, edges and weights on edges, a graph ...'.
> We also add the sentence 'For a chain of graph $\mathcal{G}$ can be created by any clustering method.' immediately before the last sentence of the paragraph.
> We have mentioned some effective clustering algorithms in the paragraph '``Chain of coarse-grained graphs for pooling' in Section 3.
>
> 9. page 5, section 3.1, Orthogonality: Define $l_2(\mathcal{G}_j)$ (I can imagine its definition, though).
>
> We added this definition, and changed the first sentence of the 'Orthogonality' paragraph to 'For each level $j=0,\dots,J$, the sequence $\{\phi_{\ell}^{(j)}\}_{\ell=1}^{N_j}$ is an orthonormal basis for $l_2(\mathcal{G}_j)$ (which is the space of all square-summable sequences on the graph $\mathcal{G}_j$) with ...'.

---

> > ### Author Response · Authors · 2019-11-13
> > **Responses to Reviewer #1 (Part 2 of 2)**
> >
> > 10. page 5, section 3.1, Haar basis: For two consective layers $j,j+1$ , ... $\to$ I had a little difficulty in understanding this sentence. Could you reconsider the wording?
> >
> > Yes. We have changed the sentence as 'Suppose two consecutive layers $j,j+1$. The first $N_{j+1}$ members of $\phi_{\ell}^{(j)}$, $\ell=1,\dots,N_{j+1}$, are defined on the finer layer $j+1$, and can be reduced into the $\phi_{\ell}^{(j+1)}$, $\ell=1,\dots,N_{j+1}$, as follows. For first $\ell=1,\dots,N_{j+1}$, $\phi_{\ell}^{(j)}(v)=\phi_{\ell}^{(j+1)}({Pa}_\mathcal{G}(v))/\sqrt{|{Pa}_\mathcal{G}(v)|}$, i.e. the value of the $\phi_{\ell}^{(j)}(v)$ is equal to the scaled $\phi_{\ell}^{(j+1)}$ at the parent ${Pa}_\mathcal{G}(v)$ of $v$ and the scaled factor is the one on square root of the number of nodes in the cluster which $v$ lies in.'.
> >
> > 11. page 13, Appendix B, (10): I think "1" in this equation is an all-one vector. It is better to write differently the all-one vector and the scalar one.
> >
> > Thanks. In (10), the '1' is the scalar one.
> >
> > 12. page 5, section 3.1 and page 7, (8): For the node $v$ in the $j$-th graph and $v'$ in the $(j+1)$-th graph, the authors use both of $v\in v'$ (section 3.1)
> > and $v'\in {Pa}_\mathcal{G}(v)$ (equation (8)) to denote the parent-child relationship. It is better to align the notation.
> >
> > We now consistently use the notation in Section 3.1 (${Pa}_\mathcal{G}(v)$) in the paper. The changes include (8) and the lines 2 and 4 in the equation of the proof of Theorem 3.1 in Appendix D.
> >
> > 13. page 13, Appendix B: Could you write in the appendix how to construct the compressed Haar from the full counterpart (I suppose we should choose bases corresponding to $k=1$)?
> >
> > The compressive Haar basis (and then the resulting compressive Haar transforms) for layer $j$ (which full Haar basis has $N_j$ vectors) is to take the first $N_{j+1}$ vectors of full Haar basis. We add a paragraph in the end of Appendix B to explain the construction of compressive Haar basis.
> >
> > Questions
> >
> > 14. In the paragraph starting with Locality on page 5, the paper says that the number of different values of the Haar basis is bounded by constant. I want the authors to make this sentence more precise. To be more specific, what "the different values" means (I imagine that it means the number of scalar values appearing as a component of some basis) and what "constant" means (with respect to which variable the "constant" is constant?)
> >
> > Yes. The 'different number of values' means the number of different scalar values of components in a basis vector. The 'constant' is independent of layers $j$. We changed last sentence of the 'Locality' paragraph as '... the number of different scalar values of the components of a Haar basis vector $\phi_{\ell}^{(j)}$, $\ell=1,\dots,N_j$, is bounded by a constant independent of $j$.'.
> >
> > 15. I think that the complexity of the direct matrix product is $\mathcal{O}(N^2 C)$, where $N$ is the graph size and $C$ is the channel size (correct me if I am wrong). Therefore the result of Figure 5 was surprising to me because the direct matrix product takes approximately time. These look inconsistent to me unless the channel size is proportional to the node size.
> >
> > Yes. The algorithmic complexity for matrix product is $\mathcal{O}(N^2 C)$. We updated the fitting for the data for matrix product (previously we used only late half of the data samples). The resulting order is $\mathcal{O}(N^{2.1})$ which is consistent to theoretical analysis as you mentioned. The figure is now in loglog plot.
> >
> > 16. Equation (5) claimed that the compressive Haar basis approximately keeps the length of an input signal. I want to know how we can justify it (theoretically or empirically). I think that it is empirically correct for the Fourier basis that signal carries 'information' in the low-frequency domains and that high-frequency areas are noisy (e.g., NT and Maehara (2019); Ma et al., 2019). If this is the case for Haar transform, too, I understand that equation (5) holds and gives the justification that the proposed method can extract the information from lower frequencies.
> >
> > By equation (5), we mean that the information in the compressive Haar transforms is close to that in the full Haar transforms for a graph signal. It does not imply the length of an input signal is preserved. In the Fourier domain, the length of the signal presentation is truncated and only low frequency coefficients are preserved. We discard high frequency coefficients but the resulting truncated presentation of the data is still close to the input signal as high frequencies carry the details of the data in the Haar representation.

---

> > > ### Comment · AnonReviewer1 · 2019-11-14
> > > **Response to authors' response**
> > >
> > > Thank you for taking my comments seriously and making efforts for improving the paper.
> > >
> > >
> > > Response to the answer to Question 16
> > >
> > > > It does not imply the length of an input signal is preserved.
> > >
> > > I am sorry for confusing the authors. By writing "the compressive Haar basis approximately keeps the length of an input signal", I intended to mean the difference between full and compressive Haar basis (i.e., $|\Phi^T_j X^{in}_j| \approx $ |\tilde{\Phi}^T_j X^{in}_j|).
> > > I understand that we want to expect that (5) holds. However, if I do not miss something, there are neither empirical nor theoretical justifications. For example, if $X_j^{in}$ is orthogonal to the compressive Haar basis $\Phi_j$, the left-hand side of (5) is $0$ and hence (5) is not true. Therefore, the authors should use some assumptions (either about Haar basis or data distribution) implicitly. I wanted the authors to clarify them.
> > >
> > >
> > > Minor comments
> > > - It makes it easier to read if the authors add to Section 3 that the authors defer the detail definition of the Haar basis to the later section (i.e., Section 4).
> > > -  Does the authors intentionally exclude the case of $N_j = N_{j+1}$ (e.g., in Definition 1)?

---

> > > > ### Author Response · Authors · 2019-11-15
> > > > **Responses to Reviewer #1, Question 16 and Minor comments**
> > > >
> > > > >I am sorry for confusing the authors. By writing "the compressive Haar basis approximately keeps the length of an input signal", I intended to mean the difference between full and compressive Haar basis (i.e.,  $|{\Phi}^T_j X^{\rm in}_j|\approx|\tilde{\Phi}^T_j X^{\rm in}_j|$). I understand that we want to expect that (5) holds. However, if I do not miss something, there are neither empirical nor theoretical justifications. For example, if  is orthogonal to the compressive Haar basis , the left-hand side of (5) is  and hence (5) is not true. Therefore, the authors should use some assumptions (either about Haar basis or data distribution) implicitly. I wanted the authors to clarify them.
> > > >
> > > > Thank you for pointing out this problem. It is a very good question, and touches the essence of Haar basis construction. We apologize for the ambiguous interpretation. We have updated the formula (5) and changed the interpretation. From construction of Haar basis, in layer $j$ of the chain, the first $N_{j+1}$ basis vectors can be reduced to the full basis in the coarser level $j+1$, and these vectors provide the approximate representation of signal on $\mathcal{G}_j$. This approximate information means  taking the average of the input data over the nodes in the same cluster. The compressive Haar transform can thus provide a pooling strategy. We compare it to the full Haar transform, as can be seen from the new formula (5). Some information in pooling has been lost and the pooled data has been integrated with the clustering structure. We give the proof of (5) in Appendix D in the modified version of paper.
> > > >
> > > > >Minor comments:
> > > > >It makes it easier to read if the authors add to Section 3 that the authors defer the detail definition of the Haar basis to the later section (i.e., Section 4).
> > > >
> > > > Thanks. We give an overview of the HaarPooling in Section 3 to give readers an intuitive impression of HaarPooling and compressive Haar basis. So we defer the mathematical definition of the Haar basis to Section 4.
> > > >
> > > > >Does the authors intentionally exclude the case of  (e.g., in Definition 1)?
> > > >
> > > > Yes. We require $N_j>N_{j+1}$, $j=0,\dots,K-1$. This makes sense as $N_{j+1}/N_j$ is the pooling ratio, which needs to be less than 1.

---

> > > > > ### Comment · AnonReviewer1 · 2019-11-15
> > > > > **Questions about the second eq. of (5)**
> > > > >
> > > > > Thank you for updating the content about (5). To confirm my understanding, let me give you two questions (I understand that it is possible that the authors cannot answer them because of the due date. So, it would be OK even if I cannot get answers).
> > > > >
> > > > > - I think the meaning of (5) has been changed from the previous version because (5) in the updated version no longer means that the compressive and full Haar bases change the length of an input signal differently. Is my understanding correct?
> > > > > - The second equation of (5) is a bit weird to me: since $\tilde{\Phi}_j$ is an orthogonal matrix, we should have $\|\tilde{\Phi}^T_j X\|^2 = \|f\|^2 = \sum_{p\in \mathcal{G}_{j+1}} \sum_{v \in p} f(v)^2$. The right hand side is different from $\sum_{p \in \mathcal{G}_{j+1}} |\sum_{v\in p} f(v)|^2 $ in general.
> > > > >
> > > > > I am sorry if I miss something because I made a quick assessment.

---

> > > > > > ### Author Response · Authors · 2019-11-15
> > > > > > **Reply**
> > > > > >
> > > > > > - I think the meaning of (5) has been changed from the previous version because (5) in the updated version no longer means that the compressive and full Haar bases change the length of an input signal differently. Is my understanding correct?
> > > > > >
> > > > > > Yes. We have a new interpretation.
> > > > > >
> > > > > > - The second equation of (5) is a bit weird to me: ...
> > > > > >
> > > > > > Yes. We just updated the paper, and the formula now becomes $\sum_{p\in\mathcal{G}_{j+1}}\sum_{v\in p}|f(v)|^2$. Thanks.

---

### Official Review · AnonReviewer3 · 2019-10-31
**Official Blind Review #3**

**Rating:** 3

**Review:**

The paper investigates the problem of graph classification using neural networks and suggests a hierarchical approach for constructing a feature vector describing a whole graph via the use of the compressed Haar transform. The general method utilizes a hierarchical chain of coarsened versions of the graph (group multiple nodes into a parent 'node')  where the coarsening is achieved via spectral clustering. After obtaining the graph chain, at each level the authors apply a GCN followed by a HaarPooling, namely applying a lossy Haar Transform compression to get a representation for each cluster.

Overall graph classification and regression tasks are quite important and this work provides a new way of transitioning from node based learning methods to full graph representations via hierarchical compression. Nonetheless, I find that the organization of the paper needs additional work and the experimental investigation is not sufficient and compelling enough. So I do not think the current paper is ready for publication in a top-tier venue like ICLR yet.

In particular I find the discussion of the HaarPooling step's computational performance not particularly appealing since other parts of the hierarchical approach are computationally expensive for example, the spectral clustering or GCN steps are costly. It also seems that the discussion of the paper is more on the computation and viability of the compressive Haar transform than it is about using it as a compressor as part of a larger hierarchical system.


**Experience Assessment:**

I have read many papers in this area.

**Review Assessment: Checking Correctness Of Derivations And Theory:**

I assessed the sensibility of the derivations and theory.

**Review Assessment: Checking Correctness Of Experiments:**

I assessed the sensibility of the experiments.

**Review Assessment: Thoroughness In Paper Reading:**

I read the paper at least twice and used my best judgement in assessing the paper.

---

> ### Author Response · Authors · 2019-11-13
> **Responses to Reviewer #3**
>
> 1. Overall graph classification and regression tasks are quite important and this work provides a new way of
> transitioning from node based learning methods to full graph representations via hierarchical compression. Nonetheless, I find that the organization of the paper needs additional work and the experimental investigation is not sufficient and compelling enough. So I do not think the current paper is ready for publication in a top-tier venue like ICLR yet.
>
> Thank you for the helpful comments. We have provided our responses below and changed the paper with your suggestion. In particular, we have improved the organization of the paper. The experimental results we provided are for benchmark data sets used in testing models for graph classification. As can be seen from  Table 2, these data sets are widely used in testing new pooling methods. We have compared with almost all recent pooling algorithms. In addition, we add a 10 graph classification experiment in Appendix F on Triangles data set which contains 45000 graph data samples. The GNN with HaarPooling outperforms the recent result by SAGPooling (Lee et al., 2019).
>
> 2. In particular I find the discussion of the HaarPooling step's computational performance not particularly appealing since other parts of the hierarchical approach are computationally expensive for example, the spectral clustering or GCN steps are costly.
>
> We focus on the computational cost of HaarPooling only as graph convolution other than GCN can be used. In Appendix F, we use GNN equipped with GIN (Xu et al., 2018) and HaarPooling on a 10 graph classification task. In addition, in Appendix G, we add Table 6 which compares the time and space complexities and other properties of HaarPooling and other pooling methods. From the table, HaarPooling is the only one which time complexity is proportional to the number of nodes.
>
> Spectral clustering is one choice. Other clustering methods can replace spectral clustering. It is adopted because it usually has better performance especially there are isolated nodes in graph. The spectral clustering per se can also be accelerated by using compressive spectral clustering algorithm, see Tremblay et al. (2016).
>
> 3. It also seems that the discussion of the paper is more on the computation and viability of the compressive Haar transform than it is about using it as a compressor as part of a larger hierarchical system.
>
> We have compressed the paper and the discussion fast computation has been reduced a bit, which takes up around one and half pages. The paper is composed of the idea, modelling, computational strategy and experiments for HaarPooling. The fast computation is only one part.

---

### Official Review · AnonReviewer4 · 2019-11-01
**Official Blind Review #4**

**Rating:** 3

**Review:**

This paper presents a new graph pooling method, called HaarPooling. HaarPooling has a mathematically formalism derived from compressive Haar transforms. HaarPooling takes into account both graph the structure and features of the graph-structured input data to compute a coarsened representation. HaarPooling can be applied in conjunction with any type of graph convolution in GNNs. Experimental results verified the efficacy of the proposed method.

The writing, organization and presentation are satisfactory.

My comments regarding this paper are as below.
1) More experiments on real tasks (e.g., multi-human parsing or pose estimation, etc.), both quantitatively and qualitatively, should be supplemented to further verify the superiority claimed in this paper.
2) The main contributions of this paper are not clear to me, compared with other SOTAs.

**Experience Assessment:**

I have published in this field for several years.

**Review Assessment: Checking Correctness Of Derivations And Theory:**

I carefully checked the derivations and theory.

**Review Assessment: Checking Correctness Of Experiments:**

I carefully checked the experiments.

**Review Assessment: Thoroughness In Paper Reading:**

I read the paper thoroughly.

---

> ### Author Response · Authors · 2019-11-13
> **Responses to Reviewer #4**
>
> Thank you for the helpful comments. We have provided our responses below and changed the paper with your suggestion.
>
> 1. More experiments on real tasks (e.g., multi-human parsing or pose estimation, etc.), both quantitatively and qualitatively, should be supplemented to further verify the superiority claimed in this paper.
>
> The data sets we use are widely used in testing pooling methods as adopted by many recent new pooling models. To make the experiments richer, we add a 10 graph classification experiment in Appendix F on Triangles data set which contains 45000 graph data samples. The GNN with HaarPooling outperforms the previous result by SAGPooling.
>
> The tasks (such as multi-human parsing or pose estimation) the referee mentioned are not typical graph classification problems. These applications are important in practice but applications are not the focus on of our paper. We focus on the development of HaarPooling model and its algorithm.
>
> 2. The main contributions of this paper are not clear to me, compared with other SOTAs.
>
> In the paper, we propose a new pooling method by using wavelet representation of the graph data. Compared with existing pooling methods, HaarPooling takes account of both node information and graph structure of the data, and it has fast implementation. In particular, HaarPooling has the following features.
> 1. This makes GNNs possible to deal with input graph-structured data with different size and structure;
> 2. HaarPooling uses the sparse Haar representation on chain structure. In each HaarPooling layer, the representation then combines the features of input $X_j^{\rm in}$ with the geometric information of the graphs of the $j$th and $(j+1)$th layers of the chain;
> 3. HaarPooling drops the high frequency (or detailed) information of the input data. The major data information (i.e. the low frequency coefficients) is preserved in the pooling, and the loss of the information is small;
> 4. The HaarPooling has near linear computational complexity as the compressive Haar basis is highly sparse. These were verified and demonstrated by numerical experiments.

---

### Official Review · AnonReviewer2 · 2019-11-01
**Official Blind Review #2**

**Rating:** 6

**Review:**

The paper presents a new approach called HaarPooling in the context of Deep Graph Neural Networks. The approach solves dmiensional problems of applying the same model on graphs of different size and shows how to contribute to high performance in a set of graph classification tasks, while having low computational complexity.

The paper is very well written. The authors describe the concept of HaarPooling in a very detailed way and show the mathematical foundation of their approach. They further provide detailed mathematical explanations and proofs for the claimed advantages of their method, whilst using descriptive examples to ease understanding. The use of HaarPooling is then tested on five different datasets, showing a very good performance on each of them. The description of the machine learning experiments is very detailed, i.e., the methodology seems reproducable.

Minor Comments:
- In Figure 3 it seems counter-intuitive that the Graphs are arranged down-top (0-->2), whilst the equations above are arranged top-down.
- The Related Work section could be placed earlier in the paper to get a better overview of the context and the problems that HaarPooling tries to solve.
- All the experiments use exactly one HaarPool layer. Would it be possible to use multiple HaarPool layers, or is this not sensible or even possible (e.g. due to the dimensionality reduction). An explanation on this could be beneficial.
- In Appendix A, weights are described with a capital 'W', which seems inconsistent with the lower case 'w' used in the rest of the paper.


**Experience Assessment:**

I have read many papers in this area.

**Review Assessment: Checking Correctness Of Derivations And Theory:**

I assessed the sensibility of the derivations and theory.

**Review Assessment: Checking Correctness Of Experiments:**

I assessed the sensibility of the experiments.

**Review Assessment: Thoroughness In Paper Reading:**

I read the paper at least twice and used my best judgement in assessing the paper.

---

> ### Author Response · Authors · 2019-11-13
> **Responses to Reviewer #2**
>
> Thank you for your helpful comments. We have provided our responses below and changed the paper with your suggestion.
>
> 1. In Figure 3 it seems counter-intuitive that the Graphs are arranged down-top ($0\to2$), whilst the equations above are arranged top-down.
>
> The figure and the above equation are consistent. The $0$ indicates the finest (or bottom) layer and the $2$ indicates the coarsest (or top) layer.
>
> 2. The Related Work section could be placed earlier in the paper to get a better overview of the context and the
> problems that HaarPooling tries to solve.
>
> We have moved the 'Related work' section forward to Section 2 so the readers will see the overview after the introduction.
>
> 3. All the experiments use exactly one HaarPool layer. Would it be possible to use multiple HaarPool layers, or is this not sensible or even possible (e.g. due to the dimensionality reduction). An explanation on this could be beneficial.
>
> For 'Protein data set', we use 3 HaarPooling layers as indicated in Table~3 in the appendix. We add a new experiment in the Section F of the Appendix which compares a two-layer HaarPooling and one-layer HaarPooling strategies. It turns out the two-layer HaarPooling has slightly higher accuracy.
>
> 4. In Appendix A, weights are described with a capital 'W', which seems inconsistent with the lower case 'w' used in the rest of the paper.
>
> In Appendix A, the weights $W_i$ are for edges. In the rest of the paper, we use $w_i$ for compressive Haar transforms, which are different from the weights for edges. So we use two symbols.

---

### Decision · Program_Chairs · 2019-12-19

**Decision:**

Reject

**Comment:**

This paper presents a new graph pooling method, called HaarPooling. Based on the hierarchical HaarPooling, the graph classification problems can be solved under the graph neural network framework.

One major concern of reviewers is the experiment design. Authors add a new real world dataset in revision. Another concern is computational performance. The main text did not give a comprehensive analysis and the rebuttal did not fully address these problems.

Overall, this paper presents an interesting graph pooling approach for graph classification while the presentation needs further polish. Based on the reviewers’ comments, I choose to reject the paper.